# Protocol for development and validation of a prediction model for post-induction hypotension in elderly patients undergoing non-cardiac surgery: a prospective cohort study

Jing-Hui Hu ®,[1,2] Ning Xu,[1,2] Zhen Bian,[1,2] Hai-Jing Shi,[1,2] Fu-Hai Ji,[1,2] Ke Peng ® [1,2]

J-HH and NX contributed equally.

¹Anesthesiology, First Affiliated Hospital of Soochow University, Suzhou, Jiangsu, China
²Institute of Anesthesiology, Soochow University, Suzhou, Jiangsu, China

**Correspondence to**
Dr Ke Peng;
pengke0422@163.com

## ABSTRACT

**Introduction** Post-induction hypotension (PIH) is a common event in elderly surgical patients and is associated with increased postoperative morbidity and mortality. This study aims to develop and validate a PIH prediction model for elderly patients undergoing elective non-cardiac surgery to identify potential PIH in advance and help to take preventive measures.

**Methods and analysis** A total of 938 elderly surgical patients (n=657 for development and internal validation, n=281 for temporal validation) will be continuously recruited at The First Affiliated Hospital of Soochow University in Suzhou, China. The main outcome is PIH during the first 15 min after anaesthesia induction or before skin incision (whichever occurs first). We select candidate predictors based on published literature, professional knowledge and clinical expertise. For model development, we will use the least absolute shrinkage and selection operator regression analysis and multivariable logistic regression. For internal validation, we will apply the bootstrapping technique. After model development and internal validation, temporal validation will be conducted in patients recruited in another time period. We will use the discrimination, calibration and max-rescaled Brier score in the temporal validation cohort. Furthermore, the clinical utility of the prediction model will be assessed using the decision curve analysis, and the results will be presented in a nomogram and a web-based risk calculator.

**Ethics and dissemination** Ethical approval was obtained from the Ethics Committee of the First Affiliated Hospital of Soochow University (Approval No. 2023-012). This PIH risk prediction model will be published in a peer-reviewed journal.

**Trial registration number** ChiCTR2200066201.

## STRENGTHS AND LIMITATIONS OF THIS STUDY

⇒ This is a prospective cohort study to develop and validate a post-induction hypotension (PIH) prediction model for elderly patients undergoing non-cardiac surgery.
⇒ The candidate predictors are selected based on the most recent published literature and clinical expertise, with easy availability and convenient clinical application.
⇒ This study includes the development, internal and temporal validation of the PIH prediction model.
⇒ The study is limited to anaesthesia induction period, without postoperative outcomes such as complications and mortality.
⇒ This is a single-centre study, and multicentre cohort studies are warranted for further external validation.

## INTRODUCTION

Hypotension is common among patients undergoing surgery and general anaesthesia. It is closely correlated with adverse perioperative outcomes, such as myocardial injury, myocardial infarction, acute kidney injury and death.[1 2] Accumulating evidence suggested that even short-term hypotension is hazardous to patients during non-cardiac surgery.[3 4] Approximately 50% of hypotension events occur between anaesthesia induction and surgical incision,[5 6] which is called post-induction hypotension (PIH). Elderly patients are often frail with comorbidities and low physiological reserve, making them more prone to perioperative hypotension. Compared with intraoperative hypotension related to multiple factors, PIH is mainly attributable to patients' preoperative conditions and anaesthetic medications, which can be predictable and preventable.

Advanced age, baseline blood pressure, preoperative comorbidities, and type and dose of anaesthetics are associated with PIH.[7 8] Preoperative volume status and sympathetic activity are also involved in PIH.[9 10] With further investigation, some PIH prediction models have been developed; however, they are mostly derived from retrospective studies with missing data and limited

number of factors, and some studies included patients in a single type of surgery only.[11-14] In recent years, new predictors have been reported, such as inferior vena cava-related parameters[15-17] and perfusion index.[18] Nonetheless, requiring specific equipment or techniques makes those predictors often difficult to be applied in the clinical practice.

Therefore, we designed this prospective cohort study to develop a universal and convenient PIH prediction model for elderly patients undergoing elective non-cardiac surgery. Using this model, we can identify potential hypotension events in advance and take preventive measures to improve anaesthesia management and patient outcomes.

## METHODS

### Study design

This single-centre prospective cohort study was conducted at the First Affiliated Hospital of Soochow University between December 2022 and August 2023. The protocol and results of this study will be reported following the transparent reporting of a multivariable prediction model for individual prognosis or diagnosis (TRIPOD) checklist (online supplemental file 1).[19]

### Study population

#### Inclusion criteria

1. Aged 65 or greater and both genders.
2. American Society of Anaesthesiologists (ASA) physical status classification I–III.
3. Undergoing elective non-cardiac surgery under general anaesthesia with tracheal intubation.
4. The duration of the surgery is longer than 30 min.

#### Exclusion criteria

1. Patients who have severe cardiovascular and cerebrovascular diseases, severe liver, kidney or lung dysfunction.
2. Undergoing nerve blocks, spinal or epidural anaesthesia.
3. Pre-existing tracheal intubation, tracheotomy or ≥2 attempts of intubations.
4. Refusal for participation.

### Study outcome

The study outcome is PIH during the first 15 min after anaesthesia induction or before skin incision (whichever occurs first). Preoperative baseline blood pressure will be recorded when patients are comfortably seated in the ward. During anaesthesia induction, patients will be monitored using non-invasive cuff blood pressure at 1 min interval. Hypotension is defined as either a 30% reduction in mean arterial pressure (MAP) from baseline or absolute MAP ≤65 mm Hg. Dual definitions for diagnosis of hypotension (either a relative reduction in MAP or an absolute threshold of MAP) have been used in many recent studies.[20-25]

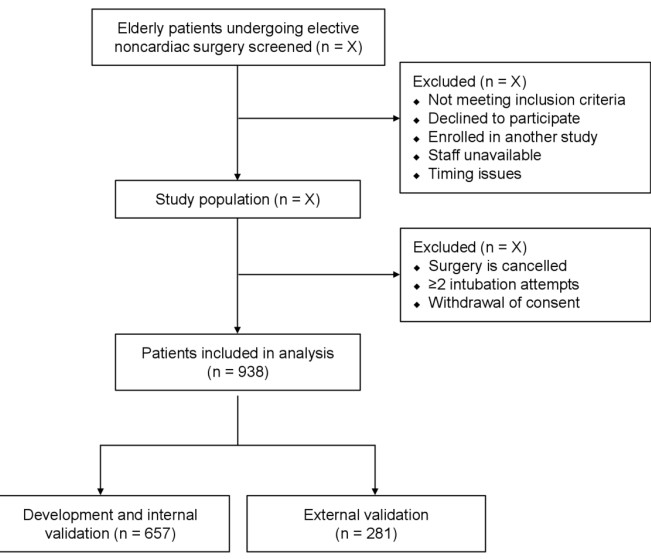

**Figure 1** Study flowchart.

## Data collection

### Study plan

One day before surgery, a research assistant will identify eligible patients and inform them about the procedures and study aims. During the study, patients can withdraw their consent at any time. If the surgery is cancelled or the exclusion criteria are met, the patient will be excluded. Figure 1 shows the study flowchart.

Candidate predictors are selected based on published data,[14] professional knowledge and clinical expertise. The following clinical variables will be recorded in the case report forms (online supplemental file 2). The research assistant will fill out the preoperative potential predictors section of the form during the preoperative visit, and document variables in the intraoperative potential predictors section and values of heart rate (HR) and non-invasive cuff blood pressure (NIBP) in the outcome section during the case when looking at the monitor.

During the analysis phase, we will calculate MAP at each time point using the traditional formula (MAP=diastolic blood pressure+0.33×pulse pressure), and we will calculate the relative threshold (30% MAP reduction from baseline). Based on these, we can determine the occurrence of PIH.

### Preoperative potential predictors

A total of 15 preoperative potential predictors:

1. Demographic data including age, gender and body mass index.
2. ASA physical status.
3. Comorbidities including hypertension, diabetes mellitus and the age-adjusted Charlson Comorbidity Index score.
4. Autonomic nervous system function (assessed by symptoms and anamneses).[26 27]
5. Myocardial injury (assessed by anamneses and preoperative examinations).
6. Vital signs in the surgical ward including MAP and HR.

7. Preoperative anxiety assessed by the first four items of the Amsterdam Preoperative Anxiety and Information Scale (APAIS).[28]
8. Preoperative frailty evaluated with the FRAIL (fatigue, resistance, ambulation, illnesses, and loss of weight) scale.[29 30]
9. Preoperative volume of fluid infusion and fasting duration.

### Intraoperative potential predictors

A total of eight intraoperative potential predictors:
1. Vital signs including MAP and HR immediately before anaesthesia induction.
2. Dose of anaesthetics used for induction, including propofol, etomidate, ciprofol, esketamine, fentanyl and sufentanil.

### Sample size estimation

According to previous literature,[5 8] the incidence of PIH was approximately 35% for patients undergoing elective non-cardiac surgery. Considering a required number of 10 events per variable,[31–33] we plan to recruit 657 patients for model development and internal validation ($657 = 23 \times 10 / 0.35$). Another 281 patients will be included continuously for model temporal validation. Therefore, the planned sample size will be set to 938 patients.

### Missing data

We will use multiple imputations when a variable has ≤5% missing data, while variables with >5% missing data will be excluded from the analysis.

### Statistical analysis

#### Descriptive analysis and comparisons

Continuous variables will be presented as mean±SD and analysed using the Student's t-test when data are normally distributed, or presented as medians with IQRs and analysed using the Mann-Whitney U test when data are not normally distributed. Categorical and ordinal data will be presented as number (%) and analysed using the $\chi^2$ or Fisher's exact tests. The absolute standardised difference (ASD) will be used to compare the baseline data between the development and validation cohort, with ASD >0.14 (ie, $1.96 \times \sqrt{(657 + 281) / (657 \times 281)}$) indicating imbalance.

#### Development of the prediction model

The development cohort (n=657) will be used to construct the model. We will use the least absolute shrinkage and selection operator (LASSO) regression analysis for variable shrinkage and selection in the development of linear regression model. For centralisation and normalisation of variables, we will use the 10-fold cross-validation in the LASSO regression analysis, which will be repeated 10 times. In the 10 times 10-fold cross-validation, we will divide the development cohort into 10 equal parts. Then 9/10th of the data will be used to retrain the model and the remaining 1/10th will be used for validating the

statistical accuracy in each time. The best lambda value will be selected. Based on the optimal predictors derived from the LASSO analysis, we will use the multivariable logistic regression analysis to develop a prediction model.

### Validation of the prediction model

The internal validation of the model will be performed using a bootstrapping method in the development cohort (n=657). A total of 500 samples will be generated and drawn randomly with replacement, and each sample will include the same number of patients in the development cohort. Moreover, the temporal validation will be conducted in the temporal validation cohort (n=281). The discrimination capability of the model will be assessed using the area under the receiver operating characteristic curve. Calibration of the model will be assessed using the Hosmer-Lemeshow goodness-of-fit statistic. The max-rescaled Brier score will be used to test the overall accuracy of the model. The clinical utility of the prediction model will be assessed using the decision curve analysis, and the results will be illustrated in a nomogram. Moreover, we will develop a web-based risk calculator by the shiny package to provide practical guide for clinic.

### Software

All data will be analysed using the R software (V.4.2.2, www.R-project.org/) and the SPSS software (V.26.0, IBM SPSS, Chicago, Illinois, USA).

### Patient and public involvement

Patients and/or the public were not involved in the design, or conduct, or reporting, or dissemination plans of this research.

## DISCUSSION

In this prospective cohort study, we will establish a model to predict PIH occurrence in elderly patients who are scheduled for elective non-cardiac surgery. Based on the main risk factors identified, the model will illustrate their effects on PIH. The effectiveness of the model will be verified via internal and external validations. To facilitate clinical application, we will visualise the prediction model in a nomogram and a web-based risk calculator. The incidence of PIH, different factors and the comparison with the existing models will be discussed in detail according to the study results.

To date, there have been many studies on the risk factors of PIH. A prospective multicenter observational study analysed 661 adult subjects who underwent elective non-cardiac surgery under general anaesthesia.[8] They found that the probability of PIH increased with age, type II diabetes mellitus, the severity of hypertension at the time of operating room arrival and the dose of propofol. In their study, the blood pressure was measured at a 5-min interval, which may result in missing of hypotension events. In our study, we set the measurement interval at every

1 min for a total duration of 15 min or before skin incision, which improves the ability to detect PIH. In addition, the baseline MAP values will be determined in the ward preoperatively to avoid possible impact by anxiety after entering the operating room.[34] Although several PIH prediction models incorporating machine learning have been reported,[11–13] their data were retrospectively collected with a limited number of patients.

In contrast, our study is based on a prospective cohort design. We select the candidate predictors based on published data, professional knowledge and clinical expertise, which ensures the feasibility and practicality in finalising the model. Regarding the APAIS score and FRAIL score, we will collect these two variables using the easy-to-use evaluation forms with good clinical practicality. Last, we will perform 10-fold cross-validation in the LASSO regression analysis and repeat it 10 times. According to the best lambda value, the optimal predictors will be derived from the LASSO analysis.[35]

## ETHICS AND DISSEMINATION

The study protocol was approved by the Ethics Committee of the First Affiliated Hospital of Soochow University on 8 November 2022, and then registered at the Chinese Clinical Trial Registry (http://www.chictr.org.cn) on 28 November 2022. The second version of study protocol was approved by the Ethics Committee (Approval No. 2023-012) on 10 January 2023 and updated at the Chinese Clinical Trial Registry on 28 January 2023. All patients will provide their written informed consent to participate in this study. All acquired data will be deidentified, stored electronically and password protected. The final prediction model will be published via a peer-reviewed journal and presented at academic meetings.

**Acknowledgements** The authors thank Shuang-jie Wu, Zhou-lin Lu and Li Zhou for their helpful collaboration in the Data Monitoring Committee.

**Contributors** J-HH, NX, ZB, H-JS, F-HJ and KP participated in the study design. J-HH, NX, ZB and H-JS drafted the manuscript. F-HJ and KP critically revised the manuscript. J-HH and NX contributed to sample size calculation and statistic plan. J-HH, NX and ZB developed the case report forms. KP is the primary investigator. J-HH, KP and F-HJ will acquire the research funding.

**Funding** This study is supported by the College Students' Extracurricular Academic Research Project of Soochow University (KY2023101A to J-HH), Jiangsu Medical Association Anaesthesia Research Project (SYH-32 021-0036 (2021031) to KP), Suzhou Medical Health Science and Technology Innovation Project (SKY2022136 to KP) and National Natural Science Foundation of China (82072130 to F-HJ).

**Competing interests** None declared.

**Patient and public involvement** Patients and/or the public were not involved in the design, or conduct, or reporting, or dissemination plans of this research.

**Patient consent for publication** Not applicable.

**Provenance and peer review** Not commissioned; externally peer reviewed.

**ORCID iDs**
Jing-Hui Hu http://orcid.org/0009-0000-9352-7714
Ke Peng http://orcid.org/0000-0003-2879-1759

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
