## [Reviewer comments · BMJ Open]

ARTICLE DETAILS

TITLE (PROVISIONAL)	Protocol for development and validation of a prediction model for post-induction hypotension in elderly patients undergoing noncardiac surgery: a prospective cohort study
AUTHORS	Hu, Jing-hui; Xu, Ning; Bian, Zhen; Shi, Hai-jing; Ji, Fu-Hai; Peng, Ke

VERSION 1 – REVIEW

REVIEWER	Si, Yanna Nanjing First Hospital, Department of Anesthesiology
REVIEW RETURNED	24-May-2023

GENERAL COMMENTS	The authors aimed to develop and validate a post-induction hypotension (PIH) prediction model for elderly patients undergoing elective noncardiac surgery to identify potential PIH in advance and help to take preventive measures. This study might be helpful in providing the optimization of anesthesia management and improving patient outcomes. There are some comments as follow. 1. I would suggest to provide the reference source for the definition of hypotension in the Study outcome section.2. Patients with Parkinson disease, as well as other central nervous system diseases, may suffer from autonomic nervous system dysfunction. These patients are susceptible to hypotension during anesthesia, as well long-term bedridden patients. I would suggest to incorporate these factors as preoperative potential predictors.3. As seen in the Introduction section, patients with myocardial injury and myocardial infarction may be subject to PIH. I would suggest to record the indicators, such as serum cardiac troponin, serum myocardial enzyme, ST-T changes from intraoperative ECG and ultrasonic cardiogram, in order to investigate whether these factors are major causes of PIH.4. It is not rigorous enough that external validation conducted in different time period of one research center in this study. I would suggest to use temporal validation, which is a method between internal and external validation.5. In Development of the prediction model section, the 10 times K cross-validation in the LASSO regression analysis was conducted only once. It's better to repeat several times to improve the accuracy of PIH prediction model.6. It has been reported that the visual methods of prediction model include figure, score or web-based risk calculator, may provide practical guide for clinic. I would suggest to clarify the visual method of PIH prediction model.
--

REVIEWER	Christensen, Anna L. Mathematica Inc
REVIEW RETURNED	I receive contract research funding from Edwards Lifesciences. 26-May-2023

GENERAL COMMENTS	This study will add to the literature on prediction of post-induction hypotension (PIH) because it is one of the first prospective cohort studies used for development of a predictive model of PIH. The authors note that it will not have the level of missing data as in previous models developed using retrospective data. The protocol is clear and logical. The authors note that the study will be conducted between December 2022 and August 2023; therefore, the reviewer assumes that data collection is complete or nearly complete, and this review focuses on requests for minor clarifications or rationale. The authors can strengthen the protocol by adding some additional detail as follows.  1. Study outcome section: Please provide rationale and citations for the dual definition of PIH you have selected. What is the rationale for defining PIH as 30% reduction in MAP from baseline or MAP < 65 mmHg, rather than selecting one definition (relative or absolute decline)? 2. In the Data Collection section, please describe how the outcomes are collected. For example,  a. The CRF outcomes section includes many fields, with 4 items to collect every minute. Please clarify who is completing the form (anesthesia clinician or study coordinator/research assistant) and when (during the case when looking at the monitor, or after the case when looking at the vitals recorded in the AIMS/EHR). This has implications for the accuracy of the outcomes data. b. Clarify if the MAP is directly available from the noninvasive cuff, or if the study team calculates MAP later, during the analysis phase, using SBP/DBP. c. Clarify if the PIH column in the CRF is intended to be recorded during the case, or if that data element is calculated later during the analysis phase. The “30% reduction from baseline” component of the PIH definition should be calculated by software during the analysis phase for higher accuracy. 3. Sample size estimation: Provide a citation or rationale for the statement: “required number of 10 events per variable.” 4. Development of the prediction model section:  a. Restate which sample the development happens on. Presumably it is the 657 patients set aside for model development (described in the sample size paragraph), but it would help with clarity to restate it in this section. b. Please elaborate on how you will decide whether a predictor variable will be included in the final model. Will you consider feasibility/practicality in finalizing the model? I highly appreciated that in the Introduction section, you mentioned the of practicality of predictor variables (i.e., it is not practical to include inferior vena cava related predictors in predictive models that will be used in routine practice.) If there are covariates that are only moderate or low predictors of PIH but require extra data collection from the patient or provider (e.g. the APAIS score or FRAIL score), will you consider leaving them out of the model? 5. Validation of the prediction model section: Clarify the population being used for internal validation, and how many are selected in each bootstrapping run. Presumably you use the development sample and select 657 cases (with replacement) but please state more clearly. Similarly, restate the population
---

	external validation is being done on. Presumably it's the 281 patients set aside for validation; however, it can be stated more clearly. 6. Figure 1: In the second box down, what are the "Other reasons" for exclusion? Please replace "other" with the specific exclusion criteria.
--	--

VERSION 1 – AUTHOR RESPONSE

Reviewer 1

Comment 1: I would suggest to provide the reference source for the definition of hypotension in the Study outcome section.

Response: Thank you for your suggestions. We have added the references in the Study outcome section. "Hypotension is defined as either a 30% reduction in mean arterial pressure (MAP) from baseline or absolute MAP \leq 65 mmHg. Dual definitions for diagnosis of hypotension (either a relative reduction in MAP or an absolute threshold of MAP) have been used in many recent studies.20-25" (See paragraph 4 in page 5, paragraph 1 in page 6)

Comment 2: Patients with Parkinson disease, as well as other central nervous system diseases, may suffer from autonomic nervous system dysfunction. These patients are susceptible to hypotension during anesthesia, as well long-term bedridden patients. I would suggest to incorporate these factors as preoperative potential predictors.

Response: Thank you. We agree with you that autonomic nervous system function is indeed a potential predictor. We have included this point in the Preoperative potential predictors section "Autonomic nervous system function (assessed by symptoms and anamneses)." (See paragraph 3 in page 6)

We also added this parameter with specific assessments in the CRF (See pages 1 and 3 in Supplemental file 2).

Comment 3: As see in the Introduction section, patients with myocardial injury and myocardial infarction may subject to PIH. I would suggest to record the indicators, such as serum cardiac troponin, serum myocardial enzyme, ST-T changes from intraoperative ECG and ultrasonic cardiogram, in order to investigate whether these factors are major causes of PIH.

Response: Thank you. Considering that these factors you mentioned are relevant with preoperative myocardial injury, we have included it in the Preoperative potential predictors section. "Myocardial injury (assessed by anamneses and preoperative examinations)." (See paragraph 3 in page 6)

We also added this parameter with specific assessments in the CRF (See pages 1 and 3 in Supplemental file 2).

Comment 4: It is not rigorous enough that external validation conducted in different time period of one research center in this study. I would suggest to use temporal validation, which is a method between internal and external validation.

Response: Thank you for your suggestions. We agree with you that using "temporal validation" is more rigorous for our study. We have changed the statement in relevant sections with "temporal validation" (See paragraph 2 in page 2, paragraph 2 in page 7, paragraph 2 in page 8)

Comment 5: In Development of the prediction model section, the 10 times K cross-validation in the LASSO regression analysis was conducted only once. It's better to repeat several times to improve the accuracy of PIH prediction model.

Response: Thank you for your suggestion. We have clarified that the 10 times K cross-validation in the LASSO regression analysis will be repeated 10 times, “For centralization and normalization of variables, we will use the 10-fold cross-validation in the LASSO regression analysis, which will be repeated 10 times. In the 10 times 10-fold cross-validation, we will divide the development cohort into 10 equal parts. Then 9/10th of the data will be used to retrain the model and the remaining 1/10th will be used for validating the statistical accuracy in each time.” (See paragraph 1 in page 8)

Comment 6: It has been reported that the visual methods of prediction model include figure, score or web-based risk calculator, may provide practical guide for clinic. I would suggest to clarify the visual method of PIH prediction model.

Response: Thank you. We agree with you that more visual methods will provide more practical guide for clinic. We will also develop a web-based risk calculator using the R software. “Moreover, we will develop a web-based risk calculator by the shiny package to provide practical guide for clinic.” (See paragraph 2 in page 8)

We also make relevant changes in the Abstract and Discussion section. “...and the results will be presented in a nomogram and a web-based risk calculator.” (See paragraph 2 in page 2)

“To facilitate clinical application, we will visualize the prediction model in a nomogram and a web-based risk calculator.” (See paragraph 1 in page 10)

Reviewer 2:

This study will add to the literature on prediction of post-induction hypotension (PIH) because it is one of the first prospective cohort studies used for development of a predictive model of PIH. The authors note that it will not have the level of missing data as in previous models developed using retrospective data.

The protocol is clear and logical. The authors note that the study will be conducted between December 2022 and August 2023; therefore, the reviewer assumes that data collection is complete or nearly complete, and this review focuses on requests for minor clarifications or rationale. The authors can strengthen the protocol by adding some additional detail as follows.

Comment 1: Study outcome section: Please provide rationale and citations for the dual definition of PIH you have selected. What is the rationale for defining PIH as 30% reduction in MAP from baseline or MAP<65mmHg, rather than selecting one definition (relative or absolute decline)?

Response: Thank you for your time and suggestions. Both absolute thresholds (e.g., mean pressure <65 mmHg) and relative thresholds (e.g., 30% reduction from baseline) are acceptable approaches to guiding intraoperative pressure management. To minimize the possibility of missing a PIH event, we will use dual definitions for diagnosis of hypotension (either a relative reduction in MAP or an absolute threshold of MAP). Dual definitions for hypotension have been used in many recent studies. We have clarified this and added citations in the Study outcome section, “Hypotension is defined as either a 30% reduction in mean arterial pressure (MAP) from baseline or absolute MAP \leq 65 mmHg. Dual definitions for diagnosis of hypotension (either a relative reduction in MAP or an absolute threshold of MAP) have been used in many recent studies.²⁰⁻²⁵” (See paragraph 4 in page 5, paragraph 1 in page 6)

Comment 2: In the Data Collection section, please describe how the outcomes are collected. For example

2a: The CRF outcomes section includes many fields, with 4 items to collect every minute. Please clarify who is completing the form (anesthesia clinician or study coordinator/research assistant) and when (during the case when looking at the monitor, or after the case when looking at the vitals recorded in the AIMS/EHR). This has implications for the accuracy of the outcomes data.

Response: Thank you. We have clarified these in the Study plan section, “The research assistant will fill out the preoperative potential predictors section of the form during the preoperative visit, and

document variables in intraoperative potential predictors section and values of HR and NIBP in the outcome section during the case when looking at the monitor.” (See paragraph 2 in page 6)

2b: Clarify if the MAP is directly available from the noninvasive cuff, or if the study team calculates MAP later, during the analysis phase, using SBP/DBP.

Response: The study team calculates MAP later, during the analysis phase. We have clarified this in the Study plan section, “During the analysis phase, we will calculate MAP at each time point using the traditional formula (MAP = diastolic blood pressure + 0.33×pulse pressure), and we will calculate the relative threshold (30% MAP reduction from baseline). Based on these, we can determine the occurrence of PIH.” (See paragraph 3 in page 6)

2c: Clarify if the PIH column in the CRF is intended to be recorded during the case, or if that data element is calculated later during the analysis phase. The “30% reduction from baseline” component of the PIH definition should be calculated by software during the analysis phase for higher accuracy.

Response: The PIH column in the CRF is determined later, during the analysis phase. We have clarified this in the Study plan section, “During the analysis phase, we will calculate MAP at each time point using the traditional formula (MAP = diastolic blood pressure + 0.33×pulse pressure), and we will calculate the relative threshold (30% MAP reduction from baseline). Based on these, we can determine the occurrence of PIH.” (See paragraph 3 in page 6)

Comment 3: Sample size estimation: Provide a citation or rationale for the statement: “required number of 10 events per variable.”

Response: We have added references for sample size estimation, “Considering a required number of 10 events per variable,31-33 we plan to recruit 657 patients for model development and internal validation ($657=23\times 10/0.35$).” (See paragraph 2 in page 7)

Comment 4: Development of the prediction model section

4a: Restate which sample the development happens on. Presumably it is the 657 patients set aside for model development (described in the sample size paragraph), but it would help with clarity to restate it in this section.

Response: Thank you for your kindly reminder. We have stated this in the Development of the prediction model section, “The development cohort (n=657) will be used to construct the model.” (See paragraph 1 in page 8)

4b: Please elaborate on how you will decide whether a predictor variable will be included in the final model. Will you consider feasibility/practicality in finalizing the model? I highly appreciated that in the Introduction section, you mentioned the of practicality of predictor variables (i.e., it is not practical to include inferior vena cava related predictors in predictive models that will be used in routine practice.) If there are covariates that are only moderate or low predictors of PIH but require extra data collection from the patient or provider (e.g., the APAIS score or FRAIL score), will you consider leaving them out of the model?

Response: Thank you. First, we have clarified the use of LASSO analysis in the Methods section, “We will use the least absolute shrinkage and selection operator (LASSO) regression analysis for variable shrinkage and selection in the development of linear regression model. For centralization and normalization of variables, we will use the 10-fold cross-validation in the LASSO regression analysis, which will be repeated 10 times. In the 10 times 10-fold cross-validation, we will divide the development cohort into 10 equal parts. Then 9/10th of the data will be used to retrain the model and the remaining 1/10th will be used for validating the statistical accuracy in each time. The best lambda value will be selected. Based on the optimal predictors derived from the LASSO analysis, we will use the multivariable logistic regression analysis to develop a prediction model.” (See paragraph 1 in page 8)

Next, we have elaborated on this point in the Discussion section, “In contrast, our study is based on a prospective cohort design. We select the candidate predictors based on published data, professional knowledge, and clinical expertise, which ensures the feasibility and practicality in finalizing the model. Regarding the APAIS score and FRAIL score, we will collect these two variables using the easy-to-use evaluation forms with good clinical practicality. Last, we will perform 10-fold cross-validation in the LASSO regression analysis and repeat it 10 times. According to the best lambda value, the optimal predictors will be derived from the LASSO analysis.” (See paragraph 3 page 10)

Comment 5: Validation of the prediction model section: Clarify the population being used for internal validation, and how many are selected in each bootstrapping run. Presumably you use the development sample and select 657 cases (with replacement) but please state more clearly. Similarly, restate the population external validation is being done on. Presumably it’s the 281 patients set aside for validation; however, it can be stated more clearly.

Response: Thank you for your kindly reminder. We have clarified this in the Validation of the prediction model section, “The internal validation of the model will be performed using a bootstrapping method in the development cohort (n=657) ... Moreover, the temporal validation will be conducted in the temporal validation cohort (n=281).” (See paragraph 2 in page 8)

Comment 6: Figure 1: In the second box down, what are the “Other reasons” for exclusion? Please replace “other” with the specific exclusion criteria.

Response: Thank you. We have replaced “Other reasons” with the specific exclusion criteria “Enrolled in another study, Staff unavailable, Timing issues” in Figure 1. (See Figure 1)

VERSION 2 – REVIEW

REVIEWER	Christensen, Anna L. Mathematica Inc
REVIEW RETURNED	22-Aug-2023
GENERAL COMMENTS	I appreciate the authors' thorough revisions. They have address all of my comments, and I have no further recommendations.

VERSION 2 – AUTHOR RESPONSE